# Polycaprolactone with Glass Beads for 3D Printing Filaments

Mária Kováčová [1], Anna Vykydalová [1,2] and Zdenko Špitálský [1,*]

1    Polymer Institute, Slovak Academy of Sciences, Dúbravská cesta 9, 845 41 Bratislava, Slovakia
2    Institute of Physical Chemistry and Chemical Physics, Faculty of Chemical and Food Technology,
     Slovak University of Technology, Radlinského 9, 812 37 Bratislava, Slovakia
*    Correspondence: zdeno.spitalsky@savba.sk

**Abstract:** At present, 3D printing is experiencing a great boom. The demand for new materials for 3D printing is also related to its expansion. This paper deals with manufacturing innovative polymer composite filaments suitable for the Fused Filament Fabrication method in 3D printing. As a filler, common and uncostly glass beads were used and mixed with biocompatible and biodegradable poly (ε-caprolactone), as a polymer matrix. This material was characterized via several physical-chemical methods. The Youngs modulus was increasing by about 30% with 20% loading of glass beads, and simultaneously, brittleness and elongations were decreased. The glass beads do not affect the shore hardness of filaments. The rheological measurement confirmed the material stability in a range of temperatures 75–120 °C. The presented work aimed to prepare lightweight biocompatible, cheap material with appropriate mechanical properties, lower printing temperature, and good printing processing. We can assess that the goal was fully met, and these filaments could be used for a wide range of applications.

**Keywords:** 3D printing; poly (ε-caprolactone); glass beads; composite; biodegradable; biocompostable; cheap material

## 1. Introduction

Currently, 3D printing, or additive manufacturing, is a significant part of the industry. Up to now, the central area of usage lies in fast prototype preparation. The most popular process of 3D printing technology is Fused Filament Fabrication (FFF), also known under the trademarked term Fused Deposition Modeling (FDM), when a continuous filament of thermoplastic material is used [1]. This 3D printing method uses thermoplastic polymers, heated above their melting point, and then extruded from a nozzle on a movable printer table layer by the pressure of the filament fed into the printhead by layer. However, there are many available materials for FFF [2]; mainly, thermoplastics are used, such as polylactic acid (PLA) [3], polyethylene terephthalate (PET) [4], polyamides (PA), acrylonitrile butadiene styrene (ABS), glycol-modified poly (ethylene terephthalate) (PET-G), and many more, such as polystyrene (PS), polyethylene (PE), poly(ethylene terephthalate) (PET), polycarbonate (PC), polycaprolactone (PCL), polyetheretherketone (PEEK), and thermoplastic urethane (TPU) [1,5]. Their main advantage is the possible filling with different fillers (fillers, plasticizers, pigments, lubricants, flame retardants, stabilizers, and chemical modifiers of material properties) to obtain polymer composites with enhanced properties [6]. Using filler (mineral, natural, or synthetic) often reduces the price of end products. Their addition to the polymer matrix affects properties (mainly mechanical and thermal), as well as thermal or electrical conductivity/resistivity, depending on the filler choice and target functionality [1]. The most often used fillers in the filaments for FDM 3D printing technology can be divided into: carbon materials (carbon black, nanotubes, graphite/graphene, carbon fibers or their combination); metal powders, ceramic, glassy and fibrous fillers (renewable raw materials such as (nano)cellulose, jute, hemp, kenaf, bamboo, flax, coconut, and others) mainly used to reinforce the structure and improve

mechanical properties; mineral fillers (graphite, titanium white, metal powders, mica, talc, chalk, diatomaceous earth), characterized by thermal, chemical and UV resistance; and different biofillers (coffee grounds, wood flour). The main disadvantage is cumulating polymer waste from an unused prototype. Solving this problem can be replacing synthetic non-degradable petrochemical-based polymers with biopolymers [7]. Many biopolymers' disadvantages are their price, mechanical vulnerability, moisture absorption, thermal stability, rapid degradability, poor performance, etc. [8–10].

Poly (ε-caprolactone) (PCL) is an FDA-approved, biodegradable polymer that has been extensively investigated for use as implantable biomaterials [11]. It is a semi-crystalline linear aliphatic polyester prepared by the chemical synthesis of crude oil, although not produced from renewable raw materials and fully biodegradable [12]. PCL can be prepared through the polycondensation of a hydroxycarboxylic acid or the ring-opening polymerization ε-caprolactone when the second way is a preferred route of preparation of PCL because it gives a polymer with a higher molecular weight and a lower polydispersity. Its main advantage for 3D printing is its significantly lower melting temperature compared to PLA (60 °C for PCL vs. 180 °C for PLA), which can dramatically reduce the operating costs of 3D printing. The second advantage is its compostable recovery after the lifetime of the prototype. PCL is used in 3D printing, mainly in blends or composites for the 3D printing of tissue engineering scaffolds [13,14] or implants [15]. To enhance the properties of PCL for 3D printing, nano-zirconium dioxide powder [16], nano-hydroxyapatite [17], carbonyl iron powder [18], strontium- and cobalt-doped multi-component melt-derived bioactive glasses [19], silver nanoparticles [20], beta-tricalcium phosphate [21], etc., were used. Recently, the first applications of PCL for 4D printing (4D printing = a 3D printed object transforms itself into another structure over the influence of external energy input such as temperature, light or other environmental stimuli) were also found. In the works of Baghani [22,23], PCL with TPU was printed in a two-layer structure with suitable shape memory performance. PCL–TPU composites had more shape fixity and recovery ratio (100% in the first cycle). Those results were better than for ABS-TPU composites. Additionally, a small amount of stress relaxation was observed compared to thermoplastic shape memory polymers with printability by FDM. The material could have many applications in bio, sensor, and actuator fields due to the minimization of stress relaxation, which is the main weakness of thermoplastic shape memory polymers.

Glass beads are high-strength, low-density, hollow glass microspheres made from soda-lime borosilicate glass in a special furnace in which it is carried upwards whilst melting in a stream of hot gases, produced by an annular flame at the bottom of the furnace. Due to the surface tension, the irregular glass particles are reshaped into spheres. Since the internal part of the particle never melts the solidification step is very fast and the spheres can therefore be collected at the top of the furnace without problems of fusion or deformation [24]. They are nonporous, chemically stable, and provide excellent water and oil resistance. Strong enough to survive processing, they can be incorporated into a wide range of polymers for density reduction. Some principal effects obtained by introducing glass beads into plastics are reduction of shrinkage, improvement in abrasion resistance, compressive strength, hardness, tensile strength, modulus, and creep. In general, the improvement of strength, modulus, and creep are less than those achieved by glass fiber at a comparable level of loading. However, because of the regular shape of the spherical filler, the improvement in compressive strength is correspondingly greater, and the reinforcing effect is isotropic since filler orientation cannot take place. Improvement of modulus has been obtained for both thermoplastics and thermosets, the enhancement is restricted since filler in the form of spheres leads to the most negligible possible effect on the overall elastic properties. The tensile strength, as well as the breaking elongation, is generally reduced, and the amount of reduction is governed by particle size and surface treatment [24]. Glass beads were used in the combination with polypropylene, polyamides, resins, and others, to improve mainly their mechanical and thermal properties [25–27].

This paper described polymer composite preparation from biodegradable PCL and hollow glass beads. The main aim was to produce and characterize a composite material that could be used as a filament for the FFF 3D printing method to reduce operational costs by significantly decreasing printing temperature below 100 °C using biodegradable polymer in combination with very cheap inert filler. The suitable mechanical, thermal, and rheological properties, are expected to be able to 3D print objects without any restriction to printing or shape stability of objects.

## 2. Materials and Methods

### 2.1. Materials

The poly $\varepsilon$-caprolactone (PCL) (CAPA 6800, PCL, Mw = 80,000 g mol$^{-1}$, melting point = 60 °C, melt flow index (MFI) = 2.4 g/10 min (2.16 kg, 160 °C)), was obtained from Perstorp (Malmö, Sweden). As a filler, glass beads 3M K15 with an average size of 60 μm and density of 0.15 g cm$^{-3}$ provided by 3M Czech Republic were used. The chemical composition was followed: $SiO_2$—min. 65%, $Na_2O$—min. 14%, CaO—min. 8%, MgO—min. 2.5%, $Al_2O_3$—min. 0.5–2%, $Fe_2O_3$—min. 0.15%, others—max. 2%. In general, to use the glass beads as a filler, the silanization reaction is necessary. This was confirmed by the Fourier-transform infrared spectroscopy comparing it with the reference glass.

### 2.2. PCL Filament Preparation

The filament was prepared according to the procedure described previously [6,28]. Briefly, composite materials were prepared in an Xplore Micro Compounder twin screw-driver (Xplore Instruments BV, Sittard, The Netherlands) with a mixing chamber (volume of 15 mL) at 65 °C. The mixture of dried polymer matrix with filler was loaded at 50 rpm, subsequently, the rate was increased to 100 rpm for 15 min and the material was drained at a speed of 50 rpm in the form of the filament. Pure PCL, and PCL/glass beads were prepared in 5, 10, 20, 30, and 40 wt%.

To test the properties, the composite was compressed into the shape of a circular plate with a diameter of 20 mm on the press Fontijne Holland SRA 100ECO 225 × 320 NA (Fontijne Holland BV, Vlaardingen, The Netherlands). The form with the sample was placed between two metal plates and then into a press machine heated to 65 °C. In the press machine, the sample was allowed to be tempered at a distance of 30 mm, after 4 min and the sample was treated with a force of 100 kN for 2 min at 65 °C. Then the press plates were cooled to room temperature.

### 2.3. Density Measurement

Density measurement was performed using the pycnometer method. A sample ($m_1$), was added to the pycnometer with ethanol of known temperature and known density ($\rho_{ethanol}$) and the pycnometer filled with sample and ethanol was weighted ($m_2$). The sample density ($\rho_{sample}$) was calculated from the weight of the pycnometer filled only with ethanol ($m_3$) according to Equation (1). The measurement was repeated three times for each sample, and the resulting density was determined as the arithmetic means of each $\rho_{sample}$. The standard deviations of the measurement were on the level of $10^{-3}$, so they are not mentioned later.

$$\rho_{sample} = \frac{(m_2 - m_1) \cdot \rho_{ethanol}}{m_4 - m_3 - m_1 + m_2} \tag{1}$$

### 2.4. Tensile Properties

The mechanical properties of filaments were measured at room temperature using a Dynamometer Instron 4301 (Instron Corporation, Norwood, MA, USA) universal testing machine at a 50 mm/min deformation rate. For each sample, 10 measurements were made, calculating the arithmetic mean and standard deviation for Young's modulus (*E*), tensile stress at break ($\sigma B$), and elongation at break ($\varepsilon B$).

*2.5. Durometer Hardness Test*

Shore D hardness test of the sample pieces was performed with a 307L type D durometer from Pacific Tranducer Corp. (Federal Ave, LA, USA) at room temperature in full compliance with ASTM Specification D2240-81. The indenter pin was forced into the sample until the durometer presser foot was flush and parallel with the material surface. Ten measurements in different spots were done, and their average was taken. All measurements were taken within seconds after the foot was in firm contact with the specimen.

*2.6. Rheology*

The rheological properties were investigated using a AR 2000 torsional rheometer (TA Instruments, New Castle, DE, USA). All the experiments were performed in oscillation mode in the parallel plate geometry with a plate diameter of 20 mm. The gap between the aluminum plates was set to 1 mm. Before measuring, the samples were equilibrated at 75 °C for 1 min in a conditioning step. Subsequently, two frequency sweeps were applied to the samples at 75 °C ($\pm 2$ °C) to investigate the melt's dynamic mechanical properties and verify the sample's thermal stability. The frequencies used ranged from 100 Hz to 0.1 Hz. A strain of 0.01% was used to maintain the material deformation in the linear viscoelastic region throughout the experiment.

*2.7. Optical Microscopy*

Samples in filaments' shape were observed using a digital microscope Leica DVM6 (Leica Microsystems, Wetzlar, Germany) with PlanAPO FOV 12.55 lens and processed by LAS X software Version 3.0.8 for 2D and 3D measurement. All samples were fractured in liquid nitrogen and inserted into the plasticine, which supports observing a cross-section of the filaments.

*2.8. Water Contact Angle Measurement*

Water contact angle measurement was performed using Surface Energy Evaluation System (SEE System; Advex Instruments, s.r.o., Brno-Komín, Czech Republic) and its software on hot-pressed samples prepared for rheology. Deionized water (3 µL droplets) was used for the measurement. The measurement for each sample was repeated 10 times, and the resulting value was reported as the arithmetic means of the measurements with the standard deviation.

*2.9. Thermogravimetric Analysis*

Thermogravimetric analysis was performed in two parallel measurements. The measurement was performed at a temperature range of 25–550 °C on a Mettler Toledo TGA/SDTA 851$^e$ (Mettler-Toledo, Bratislava, Slovakia) instrument under a nitrogen atmosphere (30 mL/min) and a heating rate of 10 °C/min. Indium and aluminum were used to calibrate the temperature. The mass of the applied samples was ~2 mg. Two parallel runs were performed for each sample.

*2.10. 3D Printing*

The ring specimens with a diameter of 20 mm and high of 1 mm to test the printability of material by FDM techniques were printed with 3D printer ORIGINAL Prusa i3 MK3 (Prusa research, Prague, Czech Republic) at a nozzle temperature of 120 °C (pure PCL and composite with 30% loading of filler) or 180 °C (for the sample with the highest loading of glass beads) without heating bed and a maximum speed of 20 mm s$^{-1}$ through a 0.4 mm nozzle.

**3. Results and Discussion**

*3.1. Microscopic Observation of Filaments*

White biodegradable filaments were prepared at a very low temperature—only 65 °C, compared to the other polymers used for 3D printing. Therefore, using PCL, as a filament

for the 3D printing of prototypes saves the energies double times; the first time during filament preparation and the second time during printing. The usage of $3\times$ lower printing temperature has a significant economic impact, especially during long-time printing. The environmental benefit of the usage of PCL is its possible composting; therefore, cumulating prototypes does not create plastic waste. The final properties of PCL can be set up by incorporating fillers; in our case, we used hollow glass beads. Their incorporation inside the polymer matrix is presented in Figure 1, where cross-section optical images of the whole filament are shown, and a detailed view of glass beads inside the polymer matrix.

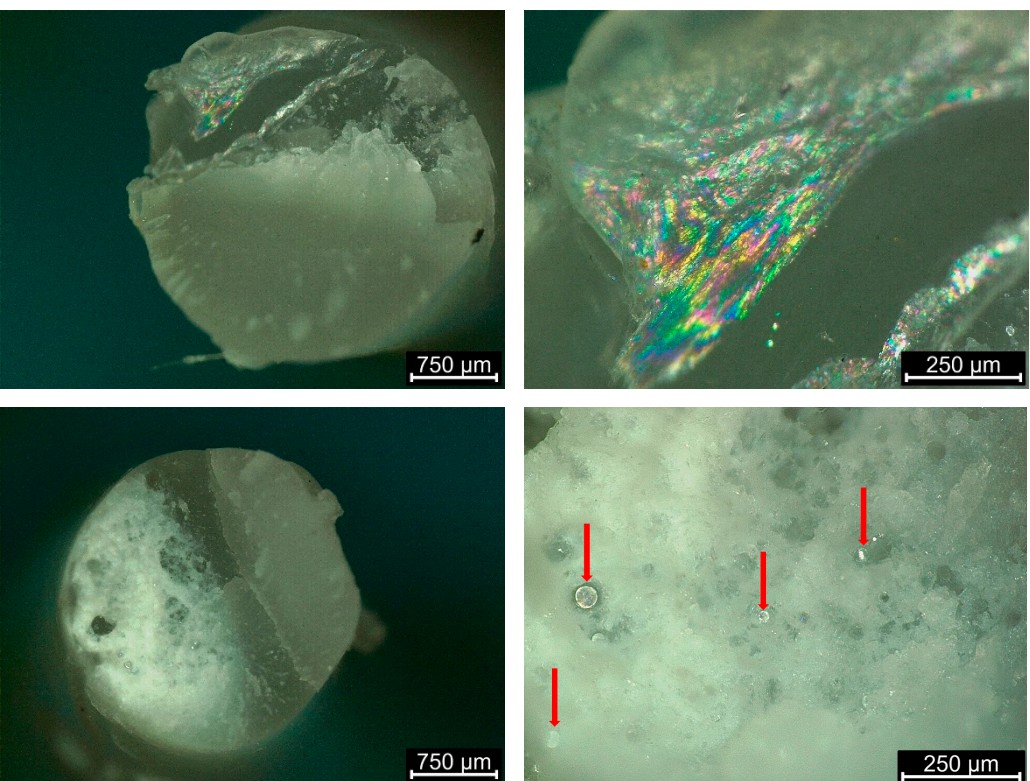

**Figure 1.** Optical images of pure PCL filament at low (**left** above) and high magnitude (**right** above). Below is a filament with 10 wt% of glass beads. The red arrow points to the placement of glass beads inside the PCL filament.

### 3.2. Water Contact Angle of Filaments

The incorporation of hollow glass beads into the PCL matrix also affects the water contact angle and density of the whole composite. What is very surprising, is that there is no linear effect with the increasing amount of glass beads. It is probably caused by the non-homogeneous distribution of filler inside the PCL matrix. However, it was not observed by optical microscopy. In any case, the composites' water contact angle is lower than pure PCL's water contact angle ($80.8°$). The lowest value was obtained for 5 wt% fillings when the value of the water contact angle was $63°$. It means the addition of hollow glass beads increased the hydrophilicity of composites. All the water contact angle results are summarized in Table 1, together with values of densities.

**Table 1.** Water contact angles and densities for PCL composites with glass beads.

| Sample | PCL | PCL5 | PCL10 | PCL20 | PCL30 | PCL40 |
|---|---|---|---|---|---|---|
| WCA (°) | $80.8 \pm 3.8$ | $62.8 \pm 2.8$ | $69.9 \pm 2.6$ | $65.5 \pm 2.4$ | $71.9 \pm 2.1$ | $67.1 \pm 4.2$ |
| Density [g cm$^{-3}$] | 1.14 | 1.15 | 1.12 | 1.18 | 1.22 | 1.30 |

### 3.3. Density of Filaments

The hollow glass beads are commonly used to decrease the model's final weight. Therefore, the effect of reducing density was also awaited in the case of PCL composites. But the opposite effect was observed. The filaments with 5 and 10 wt% loadings have the same density, but then with increasing loadings value is also increased density from $1.14$ g cm$^{-3}$ for pure PCL up to $1.30$ g cm$^{-3}$ for filament with 40 wt% loadings. There is only one logical explanation: during mixing melted polymer with glass beads inside the twin screwdriver, the glass beads at high loadings are closer together and break each other into small pieces of glass. However, this was not visible with optical microscopy due to the low resolution of the microscope.

### 3.4. Tensile and Hardness Properties of Filaments

The prepared filaments were tested for their mechanical properties, and the results are summarized in Table 2.

**Table 2.** Tensile properties and Hardness shore D values of PCL filaments with hollow glass beads.

|  | $E$ [MPa] | $\sigma_B$ [MPa] | $\varepsilon_B$ [MPa] | Shore D Hardness |
|---|---|---|---|---|
| PCL | $85.9 \pm 32.4$ | $16.5 \pm 5.0$ | $635 \pm 48$ | $56.6 \pm 2.6$ |
| PCL5 | $103.8 \pm 16.5$ | $18.7 \pm 2.5$ | $31 \pm 6$ | $57.0 \pm 0.7$ |
| PCL10 | $109.8 \pm 34.3$ | $17.2 \pm 3.2$ | $24 \pm 5$ | $56.2 \pm 0.6$ |
| PCL20 | $113.1 \pm 28.2$ | $16.7 \pm 2.3$ | $24 \pm 4$ | $55.5 \pm 0.5$ |
| PCL30 | $86.3 \pm 18.6$ | $12.9 \pm 2.6$ | $25 \pm 3$ | $54.8 \pm 1.0$ |
| PCL40 | $82.5 \pm 27.6$ | $10.5 \pm 1.4$ | $23 \pm 4$ | $54.4 \pm 0.8$ |

Note: $E$—Young's modulus; $\sigma_B$—strength at break; $\varepsilon_B$—elongation at break.

As can be seen from Table 2, the hardness of the material is almost constant and the increasing amount of glass beads does not affect its value. A different effect is observed in the case of tensile properties. The Young's modulus of pure PCL is 86 MPa and is increased with the addition of glass beads up to 20 wt% when the maximum modulus is 113 MPa (31% increase). With further loading is decreased to 83 MPa. The strength at break of pure PCL is 16 MPa, and this value is increased by adding 5 wt% of filler up to 18.7 MPa and then is linearly decreased with the increasing amount of filler up to 10.5 MPa. The elongation at break of pure PCL is 635%, but with the addition of glass beads, this value significantly decreased 20 times to 31% for 5 wt% loadings, and further loading with filler keeps this value around 24%.

The shore hardness of pure PCL is 56.6. This value is in agreement with the measurement of Luna [29] or Pavon [30], but higher than 49.8 which was observed for PCL foam [31]. It has to be noticed that PCL is a significantly softer material than the other common polymers used for FDM 3D printing, such as ABS—76 [32] or biodegradable PLA—73 [33] but close to Nylon 6-6—60 [32]. Filling the PCL with glass beads does not affect shore hardness, which remains close to the value for pure PCL. The same effect was observed on our previous filament sample of PETG filled with graphene and carbon fibers when Young's modulus and tensile stress were increased, with increasing loading. Still, nanoindentation hardness remained constant [6].

### 3.5. Rheology of Filaments

The most important parameter during the 3D printing of polymer composites by the FFF/FDM method is the flow parameter of material which rheological values can express. As mentioned above, the melting point of PCL is very low (around 60 °C), and the material is stable in a broad range of temperatures [34]. To evaluate viscoelastic parameters, the storage modulus (G′) and the lost modulus (G″) of pure polymer matrix and composites with glass beads were measured at 75 °C and 120 °C, respectively, as a function of the frequency (Figure 2). In addition, the complex viscosity ($\eta^*$) in the range of temperatures

75–150 °C for the same samples was measured. As a result, as presented in Figure 2, the material has typical thermoplastic behavior with $G''$ (open circles) over $G'$ (full circles) at lower frequencies; therefore for all samples at selected temperatures predominates liquid-like behavior. It is evident that the elastic modulus $G''$ for composites is over of $G'$ for pure polymer matrix over the whole frequency range. And this value is increasing with increasing loading of filler. As presented in Figure 2, the presence of the glass beads leads to the increase of the moduli. It means that the relaxation of polymer chains is restricted by the glass beads and the stiffness of PCL is improved. The most significant improvement is observed for the sample with 40 wt% of glass beads, when the value $G'$ is higher almost about 2 orders compared to $G'$ of pure PCL for the lower temperature and almost 3 orders for the higher temperature at frequencies below 0.1 Hz. For frequencies above 100 Hz start to appear a plateau, which means a change to solid-like behavior and stronger interactions between PCL and glass beads.

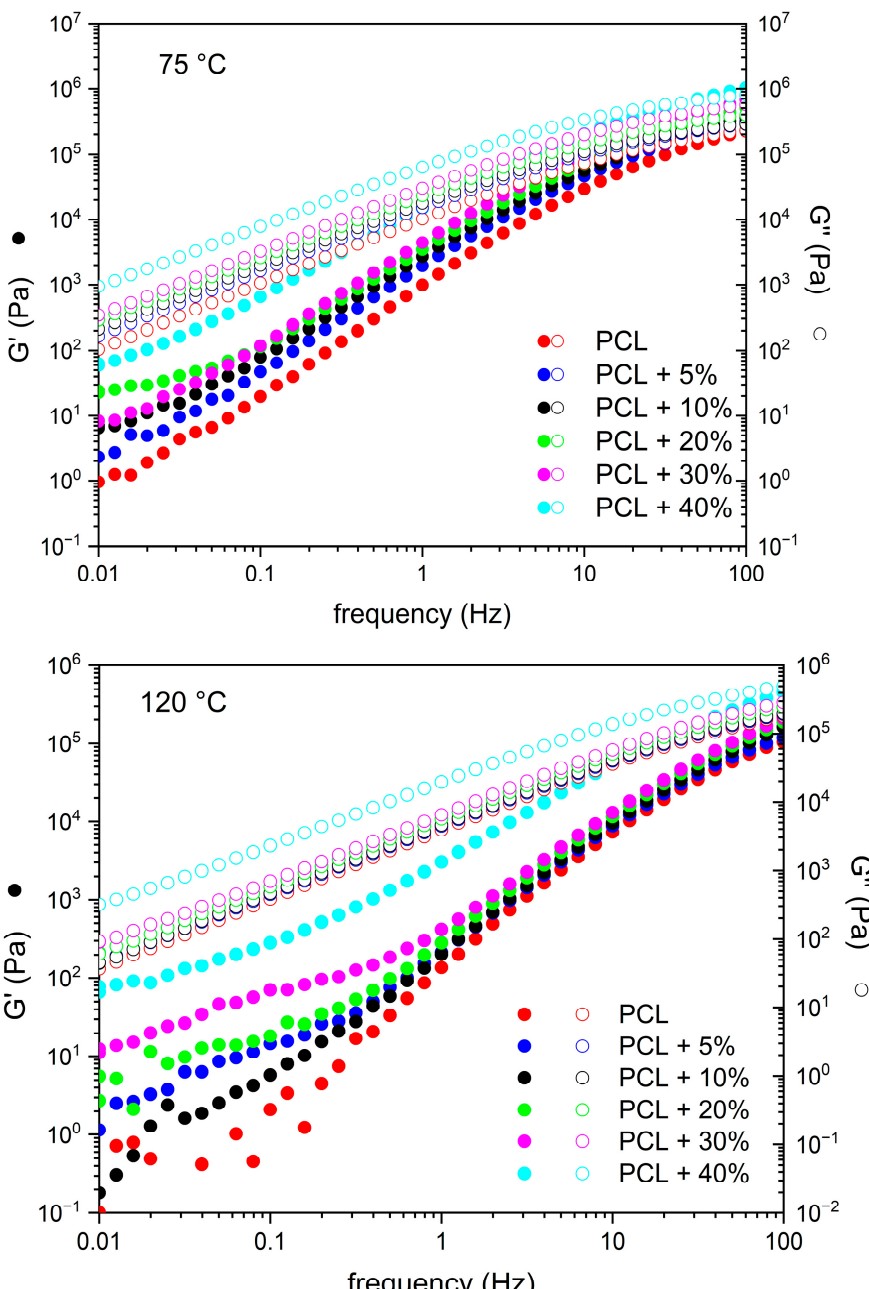

**Figure 2.** The frequency sweep of PCL and its composites with hollow glass beads at 75 °C and 120 °C.

The complex viscosity ($\eta^*$) of PCL glass beads composites is increased in comparison to pure polymer matrix in the whole temperature range. It is in agreement with the increasing of both—storage and loss moduli. The deviation of values is decreased with increasing temperature. The most significant increase in viscosity was observed for the sample of PCL with the highest loading of glass beads (40 wt%), when complex viscosity ($\eta^*$) was 6.3 times higher at 75 °C or 4.8 times higher at 150 °C, respectively. The viscosity of pure PLA pellets at 185 °C is slightly below 3000 Pa s and the material is able of 3D printing on commercial printers [28]. However, the viscosity of the material is affected by many factors during 3D printing, the complex viscosity values of pure PCL and PCL composites below 3000 Pa s are observable in Figure 3 over 90 °C for pure PCL and composites with loading up to 30 wt%. For filament with the 40 wt% loadings this value is observable only

at the temperature 120 °C and above. Based on this observation it can be concluded that all filaments should be printable on commercial 3D printers.

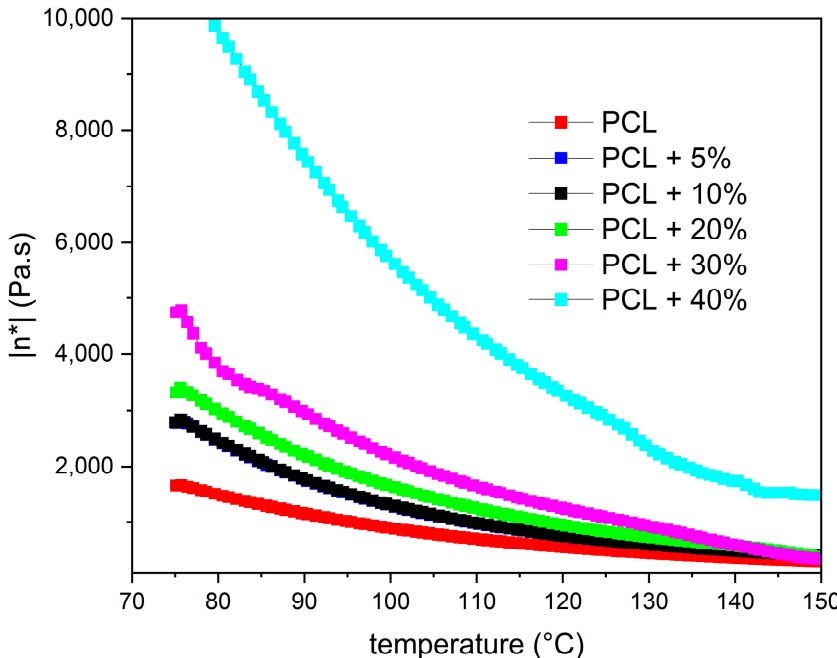

**Figure 3.** Temperature dependency of complex viscosity for PCL and its composites with hollow glass beads.

*3.6. Thermogravimetric Analysis of Filaments*

In this study, the thermal stability of all samples was carried out using the TGA analysis. The TGA curves (Figure 4) show the thermal degradation of PCL, glass beads, and their mixtures. The linear decline of mass according to weight percentage can be seen in Figure 5. This effect is also observed by TGA curves (Figure 4). The glass beads also observed the decline of their mass (around 20%) (Figure 4). We consider that more glass beads as a filler have a bad influence on the thermal stability of PCL. To a certain amount, the filler was seen as a stabilizing effect. The most significant stabilizing effect showed the addition of 5 wt% of glass beads. After the higher addition of filler, the stability of the material started to decline. The worst thermal stability showed PCL with 40 wt% of glass beads. This effect can be seen in Figures 4 and 5. We can assume that greater addition of glass beads somehow damaged the structure of PCL, so the stability started to decline. The mass at 300 °C of pure PCL has a value between PCL with 10 wt% and 20 wt% of glass beads (88.7%). A similar effect was recorded by Yuxin Zuo et al. [35]. The TGA records showed the difference between thermal degradation of 5 wt% and 10 wt% addition of hollow glass beads. They also showed the residual weight (%), which approximately corresponds to the amount of filler in the matrix (Figure 4).

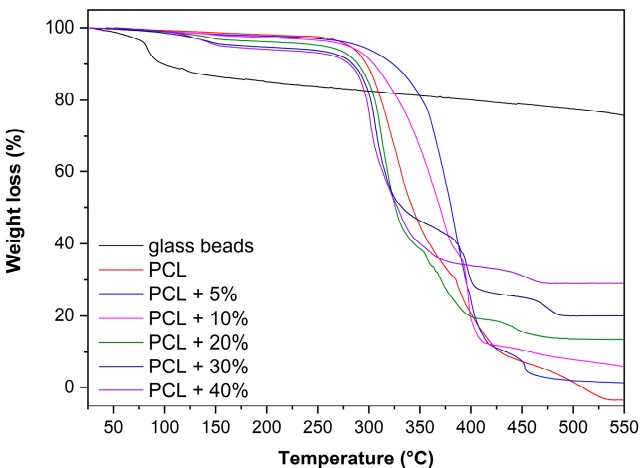

**Figure 4.** TGA records for pure PCL, glass beads, and their mixtures.

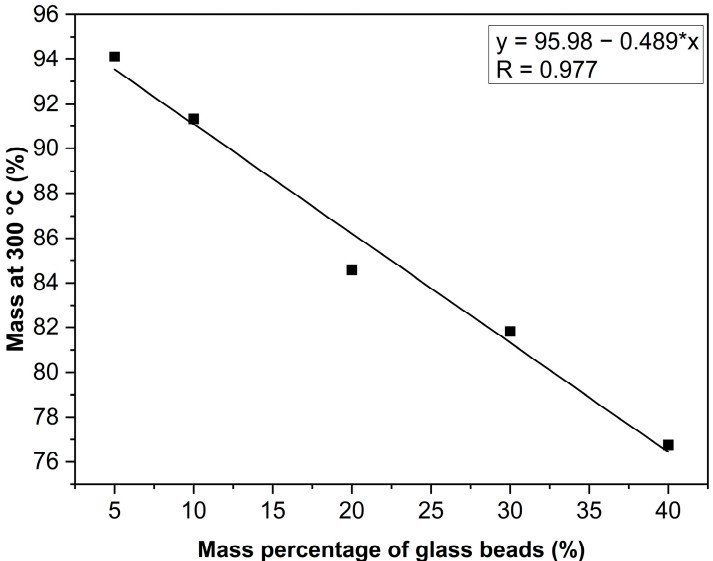

**Figure 5.** The linear regression of mass at 300 °C (%) dependence on the mass percentage of glass beads (%).

### 3.7. 3D Printing of Selected Filaments

To verify the above-described observations, three filaments were selected for printing simple circular shapes (diameter 2 cm, height 1 mm): pure PCL, PCL with 20 wt% of glass beads, and PLC with 40 wt% loading of glass beads. Pure PCL is a standard for comparison, PCL 20 is the filament with "middle" viscosity, and PCL 40 is the filament with high viscosity. All the results are presented in Figure 6.

3D printing was performed at 120 °C without heating the bed because the material was not able to solidify in the bed. This setup brings another reduction in energy consumption. The speed of printing was reduced to 20 mm s$^{-1}$ to eliminate imperfection and failures observed at higher speeds. As can be seen in Figure 6, the 3D-printed shape of pure PCL is smoothed at the surface. It is caused by very high temperature during printing. The material has a lower viscosity and flowed better on the surface at 120 °C. Therefore, it could be printed at a lower temperature than 120 °C. This temperature was used for better comparison with composites that need a higher printing temperature due to their viscosity. The PCL 20 filament can be used for 3D printing at selected conditions. The printed object is mechanically stable with characteristic roughness caused by the individual infill of the printed final layer. The last filament, PCL 40, was not able to print at 120 °C due to the clogging of the nozzle caused by the high viscosity of the composite. Therefore, the printing

temperature was increased up to 180 °C. The material was printed but the structure of the 3D shape contained a lot of failures and empty spaces. It could be enhanced by a higher temperature of printing or by decreasing the printing speed. However, this optimization will eliminate all energy savings proposed by using a low melt PCL polymer matrix.

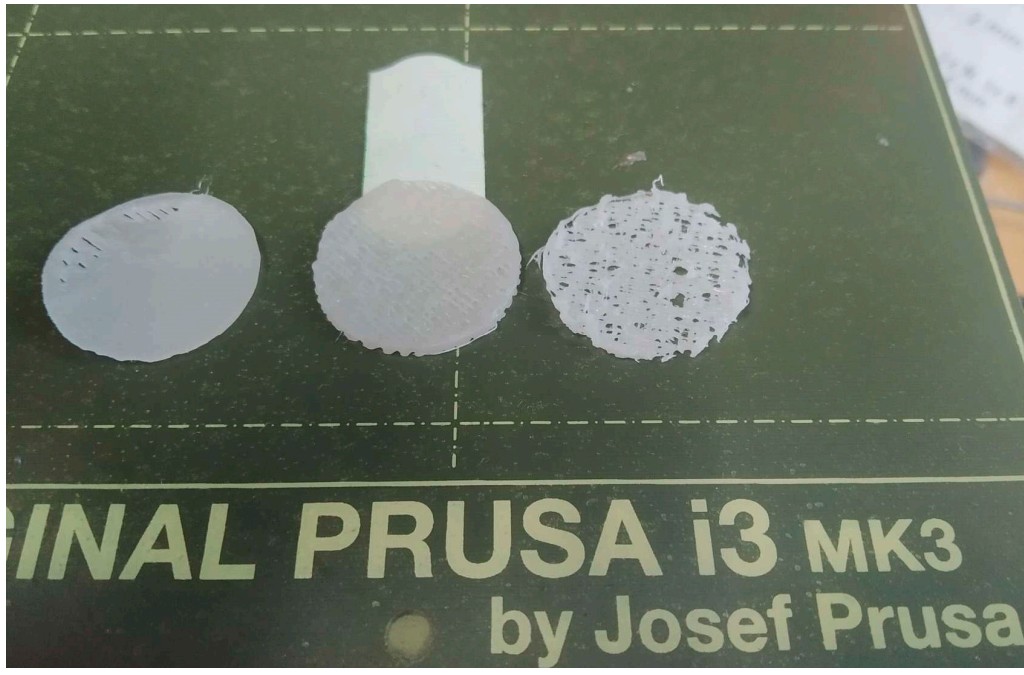

**Figure 6.** 3D printed samples of pure PCL (**left**), PCL 20 (**middle**) and PCL 40 (**right**).

## 4. Conclusions

The paper was focused on the investigation and preparation of polymer composite filaments for FFF 3D printing. Filaments were based on pure PCL, in combinations of different wt% of glass beads, as a filler. Almost all prepared filaments have a higher Young's modulus value (except 40 wt% GB), in comparison with pure PCL. Filling the PCL with glass beads does not affect shore hardness, which remains close to the value for pure PCL. Simultaneously, brittleness and elongations were decreased. The water contact angle decreased, and density increased after adding a filler. According to rheological measurement, the material is stable in a broad range of temperatures, which is very important for 3D printing method.

Moreover, a thermal degradation study showed that a high percentage of beads causes faster thermal degradation of PCL. Based on these measurements, the most acceptable sample ranges from 10 to 20 wt% GB. These composite materials have good processing properties on a 3D printer with FFF technology. The temperature of filament preparation and 3D printing was dramatically decreased. The testing of 3D printing showed that samples up to the 30 wt% loadings are printable at 120 °C; only the sample with the highest loadings was unable to print at low temperature. Thanks to this, we can evaluate that filaments bring benefits from an economic point of view (saving energies lower printing temperature, no heating of bed, uncostly filler) as well as from an environmental point of view (biodegradable/biocompatible material).

**Author Contributions:** Writing-original draft preparation, M.K.; methodology and investigation, A.V.; writing-review and editing, Z.Š.; funding acquisition, Z.Š. All authors have read and agreed to the published version of the manuscript.

**Funding:** The authors are grateful for the financial support of Grant VEGA 2/0051/20 from The Ministry of Education, Science, Research and Sport of the Slovak Republic.

**Institutional Review Board Statement:** Not applicable.

**Informed Consent Statement:** Not applicable.

**Data Availability Statement:** Not applicable.

**Acknowledgments:** The authors would like to thank our former colleague Daniela Jochec-Mošková for her help with rheology measurements.

**Conflicts of Interest:** The authors declared no conflict of interest.

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
