# Peer review of "Polycaprolactone with Glass Beads for 3D Printing Filaments"

_processes, doi:10.3390/pr11020395_

Round 1

Reviewer 1 Report

The abstract is written very briefly. Most of it contains research methods. Almost half of the abstract is devoted to generalities and research methods. The abstract should be written more attractively. Also, the novelty of the article should be presented clearly. In addition, the conducted tests and their results should be added quantitatively and qualitatively. Keywords can also be modified.

The introduction is written very superficially and briefly. Also, the sources reviewed in the introduction are very few. The introduction should be rewritten entirely. The introduction of FDM, pure thermoplastic, and composites should be added in the introduction. It is suggested to use these references, which are for new materials such as PVC, PLA-TPU and PCL-TPU multi-material printing (“Development of Pure Poly Vinyl Chloride (PVC) with Excellent 3D Printability and Macroand MicroStructural Properties” --- “3D printing of PLA-TPU with different component ratios: Fracture toughness, mechanical properties, and morphology” --- “4D Printing-Encapsulated Polycaprolactone–Thermoplastic Polyurethane with High Shape Memory Performances” --- “A New Strategy for Achieving Shape Memory Effects in 4D Printed Two-Layer Composite Structures” --- “4D printing of PET-G via FDM including tailormade excess third shape”).

The standard deviation should be added to the results in Tables 1, 2 and 3, and it does not need to be presented in a separate column.

How is the adhesion between the reinforcement and the matrix checked?

Are the evaluated results for prepared filaments or printed samples? How is the printability checked? The most important issue is choosing printing parameters, which is very challenging for new materials.

According to the rheology results, can you comment on the printability of the filament?

Most of the results sections need significant revisions, as only superficial results are reported. While each section should contain a deep discussion and analysis.

The results presented in the rheology section are general and do not need to be presented at all. More detailed information should be extracted from this test.

Figure 3 presented rheologic behavior in the temperature range of 75 – 150°C at the lowest frequency. As can be seen, the course of the temperature scan does not change with the concentration of the filler. And material became softer with increasing temperature; therefore, its printing properties can be modulated with printing temperature”

In the conclusion, the same problems as the abstract are observed.

Author Response

We thank the reviewer for the time and effort (s)he invested in the revision of our manuscript. We greatly appreciate all comments and have addressed them all. The changes in the manuscript are marked in red.

We believe that we have resolved any uncertainties and added any requested information that has improved the quality of the manuscript.

  1. The abstract is written very briefly. Most of it contains research methods. Almost half of the abstract is devoted to generalities and research methods. The abstract should be written more attractively. Also, the novelty of the article should be presented clearly. In addition, the conducted tests and their results should be added quantitatively and qualitatively. Keywords can also be modified.

The introduction is written very superficially and briefly. Also, the sources reviewed in the introduction are very few. The introduction should be rewritten entirely. The introduction of FDM, pure thermoplastic, and composites should be added in the introduction. It is suggested to use these references, which are for new materials such as PVC, PLA-TPU and PCL-TPU multi-material printing (“Development of Pure Poly Vinyl Chloride (PVC) with Excellent 3D Printability and Macro‐and Micro‐Structural Properties” --- “3D printing of PLA-TPU with different component ratios: Fracture toughness, mechanical properties, and morphology” --- “4D Printing-Encapsulated Polycaprolactone–Thermoplastic Polyurethane with High Shape Memory Performances” --- “A New Strategy for Achieving Shape Memory Effects in 4D Printed Two-Layer Composite Structures” --- “4D printing of PET-G via FDM including tailormade excess third shape”).

In the conclusion, the same problems as the abstract are observed.

We agree with the reviewer. The abstract, Introduction and Conclusion are rewritten, expanded with the requested information, and enriched by recommended references. Moreover, we added more specific keywords. These changes are not listed in the Response to referees, due to their length, but all of them are marked red in the revised manuscript.

  1. The standard deviation should be added to the results in Tables 1, 2 and 3, and it does not need to be presented in a separate column.

We merged SD with results, and Table 2 + Table 3.

Page 6, line 231:

Table 1 Water contact angles and densities for PCL composites with glass beads

Sample

PCL

PCL5

PCL10

PCL20

PCL30

PCL40

WCA (°)

80.8 ± 3.8

62.8 ± 2.8

69.9 ± 2.6

65.5 ± 2.4

71.9 ± 2.1

67.1 ± 4.2

Density [g.cm-3]

1.14

1.15

1.12

1.18

1.22

1.30

Page 6, line 248:

Table 2 Tensile properties and Hardness shore D values of PCL filaments with hollow glass beads.

E [MPa]

σB [MPa]

εB [MPa]

Shore D Hardness

PCL

85.9 ± 32.4

16.5 ± 5.0

635 ± 48

56.6 ± 2.6

PCL5

103.8 ± 16.5

18.7 ± 2.5

31± 6

57.0 ± 0.7

PCL10

109.8 ± 34.3

17.2 ± 3.2

24 ± 5

56.2 ± 0.6

PCL20

113.1 ± 28.2

16.7 ± 2.3

24 ± 4

55.5 ± 0.5

PCL30

86.3 ± 18.6

12.9 ± 2.6

25 ± 3

54.8 ± 1.0

PCL40

82.5 ± 27.6

10.5 ± 1.4

23 ± 4

54.4 ± 0.8

  1. How is the adhesion between the reinforcement and the matrix checked?

The adhesion was checked during tensile measurement, 3D printing, and microscopical observation.

  1. Are the evaluated results for prepared filaments or printed samples? How is the printability checked? The most important issue is choosing printing parameters, which is very challenging for new materials.

All measurements were performed on filaments, not on printed samples. But because of this reviewer's point, we added the chapter “3D printing” with printing parameters and also photos of the printed samples.

Page 4, line 194:

2.10. 3D printing

The ring specimens with a diameter of 20 mm and high of 1 mm to test the printability of material by FDM techniques were printed with 3D printer ORIGINAL Prusa i3 MK3 (Prusa research, Czech Republic) at a nozzle temperature of 120 °C (pure PCL and composite with 30 % loading of filler) or 180 °C (for the sample with the highest loading of glass beads) without heating bed and a maximum speed of 20 mm.s-1 through a 0.4 mm noozle.

Page 12, line 341:

3.7. 3D printing of selected filaments

To verify the above-described observations, three filaments were selected for printing simple circular shapes (diameter 2 cm, height 1 mm): pure PCL, PCL with 20 wt% of glass beads and PLC with 40 wt% loading of glass beads. Pure PCL is a standard for comparison, PCL20 is the filament with “middle” viscosity and PCL 40 is the filament with high viscosity. All the results are presented in Figure 6.

Figure 6. 3D printed samples of pure PCL (left), PCL 20 (middle) and PCL40 (right)

3D printing was performed at 120 °C without heating the bed because the material was not able to solidify in the bed. This setup brings another reduction in energy consumption. The speed of printing was reduced to 20 mm.s-1 to eliminate imperfection and failures observed at higher speeds. As can be seen in Figure 6, the 3D-printed shape of pure PCL is smoothed at the surface. It is caused by very high temperature during printing. Material has a lower viscosity and flowed better on the surface at 120 °C. Therefore, it could be printed at a lower temperature than 120 °C. This temperature was used for better comparison with composites that need a higher printing temperature due to their viscosity. PCL20 filament can be used for 3D printing at selected conditions. The printed object is mechanically stable with characteristic roughness caused by the individual infill of the printed final layer. The last filament PCL40 was not able to print at 120 °C due to the clogging of the nozzle caused by the high viscosity of the composite. Therefore, the printing temperature was increased up to 180 °C. The material was printed but the structure of the 3D shape contained a lot of failures and empty spaces. It could be enhanced by a higher temperature of printing or by decreasing the printing speed. But this optimisation will eliminate all energy savings proposed by using a low melt PCL polymer matrix. 

  1. According to the rheology results, can you comment on the printability of the filament?

The results presented in the rheology section are general and do not need to be presented at all. More detailed information should be extracted from this test. “Figure 3 presented rheologic behavior in the temperature range of 75 – 150°C at the lowest frequency. As can be seen, the course of the temperature scan does not change with the concentration of the filler. And material became softer with increasing temperature; therefore, its printing properties can be modulated with printing temperature”

We thank the reviewer for this question. We rewrote the whole rheology part, changed Figure 3, and also added part 3.7 (above mentioned).

From Page 7, line 272:

3.5.         Rheology of filaments

    The most important parameter during the 3D printing of polymer composites by the FFF/FDM method is the flow parameter of material which rheological values can express. As mentioned above, the melting point of PCL is very low (around 60°C), and the material is stable in a broad range of temperatures [34]. To evaluate viscoelastic parameters, the storage modulus (G´) and the lost modulus (G´´) of pure polymer matrix and composites with glass beads were measured at 75 °C and 120 °C, respectively, as a function of the frequency (Figure 2). In addition, the complex viscosity (η*) in the range of temperatures 75 – 150 °C for the same samples was measured.  Therefore, the frequency sweep test of all composites was performed at 75°C and 120°C, respectively. As a result, as presented in Figure 2, the material has typical thermoplastic behaviour with G´´ (open circles) over G´ (full circles) at lower frequencies; therefore for all samples at selected temperatures predominates liquid-like behaviour. It is evident that the elastic modulus G´´ for composites is over of G´ for pure polymer matrix over the whole frequency range. And this value is increasing with increasing loading of filler. As presented in Figure 2, the presence of the glass beads leads to the increase of the moduli. It means, that the relaxation of polymer chains is restricted by the glass beads and the stiffness of PCL is improved. The most significant improvement is observed for the sample with 40 %wt of glass beads, when the value G´ is higher almost about 2 orders compared to G´of pure PCL for the lower temperature and almost 3 orders for the higher temperature at frequencies below 0.1 Hz. For frequencies above 100 Hz start to appear a plateau, which means a change to solid-like behaviour and stronger interactions between PCL and glass beads.

The complex viscosity (η*) of PCL glass beads composites is increased in comparison to pure polymer matrix in the whole temperature range. It is in agreement with the increasing of both – storage and loss moduli. The deviation of values is decreased with increasing temperature. The most significant increase in viscosity was observed for the sample of PCL with the highest loading of glass beads (40 %wt), when complex viscosity (η*) was 6.3 times higher at 75 °C or 4.8 times higher at 150 °C, respectively. The viscosity of pure PLA pellets at 185 °C is slightly below 3 000 Pa.s and the material is able of 3D printing on commercial printers [28]. However, the viscosity of the material is affected by many factors during 3D printing, the complex viscosity values of pure PCL and PCL composites below 3 000 Pa.s are observable in Figure 3 over 90 °C for pure PCL and composites with loading up to 30 wt%. For filament with the 40 wt% loadings is this value observable only at the temperature 120 °C and above. Based on this observation can be concluded, that all filaments should be printable on commercial 3D printers.          

Figure 3. Temperature dependency of complex viscosity for PCL and its composites with hollow glass beads

Reviewer 2 Report

Comments to the authors

The manuscript describes the preparation and characterization of PCL composites with glass beads for using in 3D printing filaments. The authors presented the results of morphological, rheological, mechanical analysis, as well as other physicochemical properties.

I have only few suggestions:

1. Figs. 2 and 3 – I suggest to improve the quality of figures.

2. page 4, 3.1 – I suggest to add short comment about the images or to marked glass beads on the image.

3. page 6 – add more comment in the section 3.5

Author Response

We thank the reviewer for the time and effort (s)he invested in the revision of our manuscript. We greatly appreciate all comments and have addressed them all. The changes in the manuscript are marked in red.

We believe that we have resolved any uncertainties and added any requested information that has improved the quality of the manuscript.

  1. Figs. 2 and 3 – I suggest to improve the quality of figures.

We changed Figures with improved quality.

  1. page 4, 3.1 – I suggest to add short comment about the images or to marked glass beads on the image.

We marked glass beads on the image with a red arrow and added comments under the figure.

Page 5, line 215:

  1. page 6 – add more comment in the section 3.5

Thank you for the comment. We totally rewrote and extended this rheology section. 

Reviewer 3 Report

The research work is well presented and resolved. The characterization seems adequate; however, I wonder if the fracturing of the samples in liquid nitrogen may have modified their microstructure.

The prepared material is suitable for 3d printing; however, there is a significant brittleness or reduction in elongation, which I think should be highlighted and indicated in the Conclusions.

Author Response

We thank the reviewer for the time and effort (s)he invested in the revision of our manuscript. We greatly appreciate all comments and have addressed them all. The changes in the manuscript are marked in red.

We believe that we have resolved any uncertainties and added any requested information that has improved the quality of the manuscript.

The research work is well presented and resolved. The characterization seems adequate; however, I wonder if the fracturing of the samples in liquid nitrogen may have modified their microstructure.

We thank the Reviewer for pointing out the question about microstructure. Liquid nitrogen fracturing is a common procedure to observe the cross-section structure of polymers or polymer composites via microscopic observation. It prevents the stretching of the polymer on the edges of filaments. For that reason, we do not think that the given method affects the microstructure of the samples.

The prepared material is suitable for 3d printing; however, there is a significant brittleness or reduction in elongation, which I think should be highlighted and indicated in the Conclusions.

We rewrote the conclusion and added the following text:

Page 13, from line 367:

The paper was focused on the investigation and preparation of polymer composite filaments for FFF 3D printing. Filaments were based on pure PCL, in combinations of different wt% of glass beads, as a filler. Almost all prepared filaments have a higher Young`s modulus value (except 40wt% GB), in comparison with pure PCL. Filling the PCL with glass beads does not affect shore hardness, which remains close to the value for pure PCL. And simultaneously, brittleness and elongations were decreased. The water contact angle decreased, and density increased after adding a filler. According to rheological measurement, the material is stable in a broad range of temperatures, which is very important for 3D printing method.

Moreover, a thermal degradation study showed that a high percentage of beads causes faster thermal degradation of PCL. Based on these measurements, the most acceptable sample ranges from 10 to 20 wt% GB. These composite materials have good processing properties on a 3D printer with FFF technology. The temperature of filament preparation and 3D printing was dramatically decreased. Testing of 3D printing showed that samples up to the 30 wt% loadings are printable at 120 °C, only the sample with the highest loadings was unable to print at low temperature. Thanks to this, we can evaluate that filaments bring benefits from an economic point of view (saving energies lower printing temperature, no heating of bed), uncostly filler) as well as from an environmental point of view (biodegradable/biocompatible material).

Round 2

Reviewer 1 Report

Requests are well answered and considered.